# Embedding a Coaching Culture into Programmatic Assessment

Svetlana Michelle King *[ID], Lambert W. T. Schuwirth [ID] and Johanna H. Jordaan [ID]

Prideaux Discipline of Clinical Education, Flinders Health and Medical Research Institute,
College of Medicine and Public Health Flinders University, Adelaide 5001, Australia;
lambert.schuwirth@flinders.edu.au (L.W.T.S.); johanna.jordaan@flinders.edu.au (J.H.J.)
* Correspondence: svetlana.king@flinders.edu.au

**Abstract:** Educational change in higher education is challenging and complex, requiring engagement with a multitude of perspectives and contextual factors. In this paper, we present a case study based on our experiences of enacting a fundamental educational change in a medical program; namely, the steps taken in the transition to programmatic assessment. Specifically, we reflect on the successes and failures in embedding a coaching culture into programmatic assessment. To do this, we refer to the principles of programmatic assessment as they apply to this case and conclude with some key lessons that we have learnt from engaging in this change process. Fostering a culture of programmatic assessment that supports learners to thrive through coaching has required compromise and adaptability, particularly in light of the changes to teaching and learning necessitated by the global pandemic. We continue to inculcate this culture and enact the principles of programmatic assessment with a focus on continuous quality improvement.

**Keywords:** coaching; programmatic assessment; educational change; case study

## 1. Introduction

Educational change is never easy, regardless of whether the change relates to small process-related elements or a significant shift in educational philosophy that disrupts existing thinking. Small process-related changes, such as moving from a multiple-choice to open-ended question assessment, are evolutionary. More fundamental changes, such as shifting from a traditional model of assessment toward a programmatic approach, could be considered revolutionary. As would be expected, the more fundamental the change, the more complex the change management process because of the involvement of many moving parts [1,2].

Fundamental change can be prompted or initiated by a range of factors—external drivers and rationales, an internal impetus for change and innovation, the availability of staff expertise, or readiness for change—all of which are supported and/or constrained by enabling and boundary conditions. Enacting such a change is complex because of the continuous interaction between these factors [3]. It is, therefore, logical that managing a multitude of factors and their complex interactions needs to be underpinned by a supportive organizational culture that embraces innovation and change.

In higher education, enacting fundamental change is challenging. Universities are often established organizations with long-held traditions. When organizations are steeped in tradition, there can be a tendency towards a process rather than a goal orientation. To illustrate this, consider the migration from face-to-face to online lectures, often lauded as an educational innovation. Although the delivery mode differs, the educational activity essentially remains unchanged. Adopting a goal orientation would involve a redesign of education to achieve the goal (i.e., learning), rather than focusing on the process (i.e., the lecture). The redesign would then consider how learning can be optimized with technological affordances [4]. Although a focus on tradition and process orientation can enhance an organization's credibility in society, it can reduce the organization's agility

to enact fundamental change, even when it is necessary. This is where the culture of an organization becomes particularly critical.

An integral part of an organization's culture is its people. Their values, convictions, experiences, and beliefs contribute to, and are influenced by, the organizational culture. For example, people who work in universities typically hold tacit beliefs about education that are framed by the broader organizational culture. Consequently, individual beliefs can be shaped and reinforced by the organizational environment, creating a proverbial 'echo chamber'. Given the central role of an organization's people in contributing to its culture, this can have implications for educational change.

Organizational leaders that recognize these factors can enact change by identifying and responding to external drivers and societal needs to develop new insights. These three factors—external drivers, new insights, and changing societal needs—have all contributed to the implementation of programmatic assessment around the world. For instance, in our part of the world, the Australian Medical Council and the organization of Medical Deans of Australia and New Zealand suggest that the principles of programmatic assessment align with the tenets of competency-based education [5]. They also foresee a future healthcare system that provides greater patient-centered and interprofessional care, similar to the visions of other countries [6]. Yet, as we have highlighted, such fundamental change is complex and fraught with difficulty.

In this paper, we present our experience of enacting a fundamental change in our organization, by embedding a coaching culture into programmatic assessment. We make a distinction between programmatic assessment and assessment for learning. By programmatic assessment, we mean a set of principles including the development of a meaningful, shared narrative; the synthesis of assessment information derived from multiple, diverse assessment tasks [1]; and disconnecting assessment moments with decision-making and proportional decision-making. In assessments for learning, assessments are focused on directing student learning by providing meaningful feedback to the learner to generate learning plans [7], a focus on the process as well as the outcome of learning, and facilitating greater student agency in contributing to their assessment [1].

We use a linear approach to describe the introduction of coaching as an integral component of programmatic assessment, both of which were implemented simultaneously. We reflect on both the successes and challenges in making these fundamental changes based on our perceptions of the organizational culture, whilst recognizing the difficulties in critically perceiving the organizational culture from within. We conclude with some lessons we have learned from engaging in this change process.

## 2. The Case

Flinders University is a medium-sized university with a four-year graduate-entry Doctor of Medicine (MD) program. The medical program spans a large geographical footprint, with campuses in South Australia and the Northern Territory. The MD has just over 700 student enrolments across the four-year program. The MD program has strong, established relationships with the teaching hospitals in South Australia and the Northern Territory, with a large number of clinical academic status holders who also contribute to the teaching program.

Flinders University MD staff and students have diverse personal and professional backgrounds. Our medical educators have backgrounds in medicine and other health disciplines, law, and education, and our medical students' backgrounds span the social, biomedical, and clinical sciences, education, law, and engineering. The student cohort is also culturally diverse and includes international, domestic, and Indigenous students, and those from both rural and refugee backgrounds.

*The Structure of the Flinders University Medical Program*

Each year of the Flinders University MD program is divided into two semesters across the calendar year. The first two years of the MD program are predominantly pre-clinical,

with some clinical experience components, and are intended to provide students with a foundational understanding of the body's systems; an applied understanding of basic clinical skills; an opportunity to examine ethical, legal, public health, and Indigenous health issues; and a fundamental understanding of research and scholarship. Students engage in team-based learning (TBL), clinical skills simulation, practical sessions, tutorials, and lectures. Assessment involves both formative and summative tasks including TBL tests, assignments, block tests, and practical examinations. The final two years consist mainly of clinical rotations. Students can elect to undertake these placements in either an urban or rural setting. Assessments in the final two years of the MD program include workplace-based assessments including practical clinical examinations (e.g., mini-CEXs) during rotations, and a global assessment of progress at the end of each rotation.

Each semester, students are provided with a Statement of Assessment Methods that is used across the university. This document stipulates which assessment tasks are formative, and those which constitute hurdle (summative) tasks. Consistent with assessment for learning principles, students receive feedback on each assessment and are encouraged to utilize these as opportunities to evaluate their learning strategies and current knowledge and understanding ahead of engaging in summative assessments.

### 3. Programmatic Assessment: The Impetus for Change

Since its establishment in 1974, the Flinders University medical program has developed a reputation for educational innovation. Over the years, the medical course has introduced Problem-Based Learning (1995), a four-year graduate entry program (1996), and an undergraduate pathway into the MD program to fast-track high-achieving school leavers (2010). The Flinders University medical program is steeped in a culture of excellence and quality improvement, striving to ensure that the curriculum remains current and relevant to the communities it serves. This focus on meeting community needs is evident in the extension of the medical course to the Northern Territory (2011), the establishment of a parallel rural community curriculum, and the introduction of longitudinal clerkships in both rural and metropolitan settings.

From an assessment perspective, several process-related changes were made, which focused on objective measurements of end-point learning and psychometric validation of assessment outcomes. In so doing, however, it became apparent that the emphasis on measurement had come at the cost of the value of narrative evaluations of student performance. This was highlighted in student evaluation data wherein there was an expressed need for more individualized, specific, and timely feedback. At the same time, medical course staff were progressively beginning to recognize the importance of having a balanced assessment program, which recognizes the merits of feedback for learning, and the need to utilize a range of assessment outcomes to inform progression decisions. This local evaluation was complemented by the growing evidence in the literature demonstrating the negative impacts of testing regimes on students' learning, and the need to ensure that assessments were fit-for-purpose in assessing all learning outcomes [8–11]. It also became apparent that traditional educational models do not equip students to systematically reflect on their performance over time or adopt a proactive approach to their learning. Together, these internal and external drivers provided the impetus for a fundamental change in the approach to teaching and learning, giving rise to the introduction of programmatic assessment and its underpinning philosophy of assessment for learning.

### 4. Implementing Programmatic Assessment

Programmatic assessment was implemented in the Flinders University MD program in 2017 as a four-year staged process. The educational model was implemented in the Year 1 cohort in 2017, the Year 1 and 2 cohorts in 2018, the Year 1, 2 and 3 cohorts in 2019, and the Year 1–4 cohorts in 2020. Consequently, 2020 marked the first year of full implementation of programmatic assessment across the four-year MD program.

As mentioned earlier, there were several drivers for change, both within the university amongst staff and students, but also in the educational literature. These internal and external drivers for change, coupled with a need for new insights, and a recognition of changing societal needs, were focal points during the planning phase.

The initial attempt to establish a programmatic approach was made using a standard change management approach. This comprised (1) a detailed project plan; (2) the development of an implementation budget; and (3) a stepwise implementation plan. This approach proved problematic, however, for several reasons.

Organizationally, the medical course sought to introduce a longitudinal approach to curriculum and assessment in the context of a highly modularized university structure. So, a compromise had to be reached that would allow for a longitudinal approach within established university frameworks. This resulted in the development of larger, semester-based modules, equivalent to a full-time study load and comprised of several smaller university modules. This proved organizationally complex because a lack of central assessment quality control processes made it difficult to align programmatic assessment principles to university policies and procedures.

From a cultural perspective, there was no pre-existing, coordinated approach to assessment within the medical course. The lack of a unified approach to assessment resulted in extensive variation in the design and administration of assessments across the semester-based modules. Consequently, there was poor communication between staff, and between staff and students, resulting in inconsistent messaging. In turn, this poor communication contributed to resistance to change because of uncertainty, frustration, and confusion. Additionally, there was a lack of integration between the semester-based modules and the disciplines taught across the program. This created challenges for students in terms of synthesizing and utilizing information to evaluate their progress. Furthermore, because of the fragmented approach to assessment, with control residing with each individual module coordinator, the system quickly became rigid, stifling students' ability to develop holistic and meaningful learning goals.

At the same time that planning was underway to introduce programmatic assessment, a pilot portfolio and coaching program were conducted in 2016 by a small group of academic and clinical staff (*n* = 7) with Year 1 medical students (*n* = 169) across both South Australia and the Northern Territory. The pilot involved two components: (1) A reflective writing portfolio, in which students documented their progress, reflecting on their strengths and weaknesses, and identifying their learning needs to develop learning goals; and (2) individual coaching meetings with students. Following the pilot, evaluation data were obtained from pilot participants. This feedback was then used to generate recommendations relating to processes, role clarity, training, and support, which informed the implementation of these aspects of programmatic assessment.

## 5. Flinders University MD Elements of Programmatic Assessment for Learning

Three elements support the Flinders University MD program's approach to programmatic assessment: (1) progress testing; (2) the professional learning portfolio; and (3) the learning coach initiative. These three elements align with the theoretical principles of programmatic assessment in that they are longitudinal, facilitate synthesis and evaluation of assessment information from multiple sources, and provide the student with support in relation to their learning approaches.

### 5.1. Progress Testing

All medical students, irrespective of their year level, sit a Progress Test [12] four times per year throughout the four-year MD program. Each test includes a unique set of questions, but all students (irrespective of their year level) complete the same test. The Progress Test aims to support students to evaluate their knowledge development over time, as they progress through the medical program. This test is informative for students in the first two years and summative for students in the final two years. By informative, we mean

that students are required to undertake the assessment, but the Progress Test scores do not contribute to their final semester grade. For students in the final two years of the program, Progress Test results do contribute to end-of-semester progression decisions.

After each Progress Test, statistical analyses are performed and students' scores are converted to one of three qualifications (i.e., satisfactory, doubtful, or unsatisfactory). In addition, the student receives an extensive breakdown of their own scores, feedback on their performance on each question and the associated 'theme' of the question, and information about the performance of their peers. Although this is a norm-referenced assessment and may seem to contradict the tenets of programmatic assessment for learning, this approach is only used to ensure comparability of outcomes over consecutive assessments. It is, therefore, intended to ensure that students can detect meaningful longitudinal patterns in the development of their applied knowledge.

Students receive this information, together with a breakdown of the questions and their scores in relation to the cohort. This information is intended to serve as a diagnostic tool to support students to identify deficits in their knowledge and review and tailor their study strategies. Consequently, all students are expected to analyze their results after each progress test and use this feedback to evaluate their strengths and areas for development and generate learning goals and plans based on this evaluation. Students develop their own approaches to analyzing their results using, for example, graphs and tables, and presenting this in their professional learning portfolio (discussed later). To provide additional scaffolding for this process, we have recently developed and introduced a resource to support students to maximize the use of their progress test feedback to inform their learning goals and plans. This analysis and the meaning-making process are facilitated by the student's learning coach (discussed later).

Because progress testing is a high-stakes assessment in the final two years of the program where the results contribute to progression decisions, Year 3 and 4 students who receive a doubtful or unsatisfactory result are offered 'strategic study coaching' over and above the support they receive from their learning coach. Strategic study coaching provides students with an opportunity to receive targeted support from a coach with particular expertise in supporting students to evaluate their performance, use this information to review their long-term learning strategies, and make targeted improvements to their learning.

Although it is an important component of the assessment program, the progress test generates four data points each year, providing an indication of the student's learning and development. We can use the creation of an image as an analogy, wherein the progress test represents 'pixels' that contribute to the whole image (i.e., the student's development over time) [13]. Central to PAL is the learner's active role in deriving meaning from these pixels—each assessment event or parts of assessments—and the feedback this provides about their learning and development [14]. This sense-making process is facilitated by the professional learning portfolio and supported and scaffolded by the learning coach.

### 5.2. The Professional Learning Portfolio

To support students to develop their capacity for reflective practice and self-regulated learning [15], all students develop and maintain an ePortfolio in which they reflect on evidence from their learning, assessment, and feedback to evaluate their progress and development. As with the progress test, the professional learning portfolio is a core feature of the four-year program and is integrated across the curriculum.

The professional learning portfolio is structured according to the Flinders University MD program's eight course learning outcomes. These learning outcomes cover scholarship, knowledge, skills, communication, self-regulated learning, society, leadership, and professionalism, and are aligned to the graduate outcomes stipulated by the country's accrediting body, the Australian Medical Council [16]. The Flinders University MD course learning outcomes are intended to support students' development as well-rounded junior doctors who are " . . . knowledgeable, skilled and up-to-date . . . and undertake to maintain and develop their expertise over the course of a lifelong career" [17] (p. 3). Each semester,

students must demonstrate ongoing, meaningful engagement with the professional learning portfolio, evidenced by at least two entries per course learning outcome, written at various timepoints throughout the semester. Students are encouraged to reflect at a time that suits them, and to focus their attention on identifying their strengths and areas for development, goal setting, and goal review, in addition to significant learning experiences that influence their ongoing learning and development towards becoming a doctor. To support this engagement, students are provided with a rubric, which outlines the criteria for satisfactory engagement with the professional learning portfolio.

In the professional learning portfolio, students analyze their learning experiences, feedback, and assessment results to identify and evaluate their strengths and areas for development. These insights are then used to inform the development of individual learning goals and plans. Each semester, students identify and work towards achieving a learning goal for each course learning outcome, and evaluate their progress towards achieving these goals. All of these reflections, evaluations, and analyses are captured in the student's professional learning portfolio. This structure serves to establish a continuous cycle of analysis and reflection, goal setting, and evaluation in response to learning experiences [18]. In this way, the professional learning portfolio moves beyond a focus on reflection as a ritual, toward meaningful and purposeful reflection [19].

The intention of the professional learning portfolio is to support students to develop their reflective capacity for self-regulated learning, which focuses on goal setting for learning, and monitoring, regulating, and controlling cognition, motivation, and behavior [20]. These core skills are developed throughout the Flinders University MD program and are intended to support students to engage in ongoing reflective practice requirements as they commence clinical practice.

### 5.3. The Learning Coach Initiative

To support students to identify and utilize feedback and evidence from their learning, develop learning goals, and reflect upon this in their professional learning portfolio, the Flinders University MD program includes a learning coach initiative that is integrated across the four-year course. This constitutes a systematic and deliberate support mechanism that serves to enhance students' capacity to engage in reflective practice. This is aligned with educational literature that posits that engagement with portfolios and reflective learning is enhanced through mentoring [19,21].

Upon enrolment, each student is allocated a learning coach with whom they work from commencement to completion of their MD. Learning coaches support students to achieve their fullest potential by assisting them to evaluate and reflect upon their performance, identify their learning needs, and develop action plans to address these needs [22]. Coaching occurs at regular intervals (i.e., once per semester) throughout the four-year program, and takes the form of one-on-one meetings to discuss the student's progress towards internship and provide scaffolding and support to enhance the student's capacity for self-regulated learning. This includes coaching students to develop learning goals that are specific, measurable, achievable, relevant, and timely ('SMART') [23,24]. Coaching facilitates students' ability to make sense of the information (i.e., the 'pixels') they receive in relation to their learning in the broader context of their ongoing development towards professional practice (i.e., the larger 'image'). Together, this supports students to move away from focusing on learning to pass assessments, towards a focus on learning to understand and develop the knowledge and skills needed to become competent, safe medical practitioners.

To facilitate the coaching process, students develop meeting agendas and maintain formal records of their discussions with their learning coach. Templates that are structured according to the course learning outcomes, aspects that the student wishes to discuss, and students' learning goals are provided. These templates serve as a structure for learning coach meetings and are provided to enable students and learning coaches to focus on the *content* of meetings. In preparation for meetings, learning coaches review their students'

academic results, professional learning portfolio entries, and the agenda. Students then submit a written summary of the key points discussed during the meeting, which is endorsed by the learning coach. At the end of each semester, students prepare a summary that outlines their progress towards achieving their goals and provide a self-assessment of their readiness to move into the next semester of the program. For this summative component, the learning coach's role is to review this summary and provide an endorsement of the student's self-assessment.

From time to time, students may require additional support beyond the formal learning coach meeting that occurs once per semester. This support may relate to assisting students to address academic or wellbeing challenges. Where coaches have the capacity (i.e., time), willingness, and perceive that they have the necessary expertise, they can make themselves available for students to contact them in between meetings should they require further coaching. This is, however, dependent on the individual coach, given the additional investment of time required. Additionally, learning coaches and students are encouraged to contact the learning coach coordinator if further support is required that cannot be provided by the student's learning coach. Depending on the issues identified, the student may be referred to another staff member with the appropriate expertise or could take the form of an individual discussion between the student and the learning coach coordinator, or a discussion between the student, their learning coach, and the learning coach coordinator.

Learning Coach Recruitment, Training, and Support

From September each year, the learning coach team engages in a recruitment drive involving multiple methods to enlist learning coaches for the following academic year. In addition to appealing to existing learning coaches, new recruits are identified via advertisements in regular eNewsletter communication at both the MD program and College of Medicine and Public Health level. Most learning coaches, however, are identified via word-of-mouth, which supports the assessment of the individual's suitability to engage in the role.

Our learning coaches come from diverse professional backgrounds including medicine, paramedics, behavioral health, epidemiology, biotechnology, and education. Their involvement in the MD program also varies and includes core academic staff, academic status holders, guest lecturers, and clinical placement supervisors.

Following recruitment, all new learning coaches engage in an initial training program at the start of the calendar year. This involves a self-paced online module that introduces learning coaches to the MD program, its components, and the concept and theoretical underpinnings of coaching. This is followed by an interactive session to further discuss theoretical frameworks that can be used to inform coaching practices, review examples of reflective writing, and engage in case-based discussion to begin to develop coaching skills. This initial training is complemented by opportunities to calibrate expectations at each stage of the MD program, prior to the commencement of each learning coach meeting cycle.

Learning coaches also have opportunities to participate in, and lead, coaching community practice sessions. These include 'Coaching Conversations' and 'Portfolio Review Panels'. Coaching Conversations are held throughout the year and involve deidentified case-based discussions based on individual learning coaches' experiences of coaching. Portfolio Review Panels occur prior to the end of each semester and are intended to support learning coaches with their endorsement of a student's self-assessment of progress. Panel discussions involve a student case presentation by the student's learning coach. Learning coaches who are present are provided with access to the student's professional learning portfolio. In both community of practice environments, the intention is to enhance individual and collective coaching capacity. It also serves to establish a culture of sharing coaching practices and strategies.

## 6. Evaluation

Now in its sixth year of implementation, the idea of programmatic assessment has become firmly ensconced in the fabric of the Flinders University MD program. Although, from a 'purist' perspective, one might argue that full implementation of programmatic assessment remains a work-in-progress. For example, many assessments serve as summative, barrier tasks with minimal contribution to synthesis in the professional learning portfolio. This is a clear illustration of the point we made in the introduction, that educational change, when it challenges fundamental beliefs and traditions in education, is very hard to action. Academic staff often have valuable teaching experience but may not all have formal pedagogical training. It is, therefore, logical for staff to place trust in the way they were educated and regard change and innovation as difficult. The increased casualization of the workforce with high staff turnover, heavy teaching and research workloads, and change fatigue are typical examples of additional boundary conditions that we discussed at the beginning of this paper. That being said, given the leadership of our MD program coupled with the hard work and dedication of many staff members, every change that has thus far been implemented has followed the direction of a programmatic assessment for learning approach.

It is important to recognize that during the planning and initial implementation of programmatic assessment, there was a university-wide restructure. This restructure had implications for staff positions, workload, and key tasks, with staff involved in the implementation of programmatic assessment also being affected. It is, therefore, almost impossible to determine how the implementation of programmatic assessment may have impacted the university context and the extent to which the changing university context impacted our implementation of programmatic assessment. However, to claim that developments were coincidental would be wrong.

A key factor that has contributed to a culture of programmatic assessment has been a strong focus on building relationships and alliances between academic and professional staff, clinicians, and students. These relationships have led to a strong collective desire to continue to improve the Flinders University MD program, with governance and structures to support and facilitate these efforts. Early indications suggest that the Flinders University MD program's focus on improvement is translating to our medical graduates who demonstrate better reflective capacity and self-regulated learning in the clinical environment and view assessments as opportunities for learning.

To support a systematic approach to reviewing the impact of the implementation of PAL, we conducted a research and evaluation project in 2020, involving current MD students and learning coaches across the four-year program. From an evaluation perspective, we were interested in identifying the strengths and areas for development in relation to programmatic assessment and the learning coach initiative. The evaluation revealed some key successes relating to the role of the learning coach in providing an ongoing mechanism to support students to reflect on their overall progress and facilitate their capacity for self-regulated learning and reflective practice. In particular, the development of the student/learning coach relationship over time towards internship and the structure of the coaching initiative were considered key strengths. The evaluation also revealed aspects for further refinement of the program, predominantly reviewing our processes to reduce their complexity (e.g., grading of tasks associated with meetings, and navigating multiple systems in preparing for meetings) and further enhance support to better understand self-regulated learning. We continue to undertake research and scholarship in coaching and reflective practice to not only enhance the program at a local level but also contribute to the health professions education community.

We cannot stress enough the importance of having a small team of dedicated staff who acted as knowledge brokers during the planning and implementation phases. These staff had a clear vision of what programmatic assessment would look like at Flinders University, which was coupled with an agile approach to implementation, involving a combination of goal orientation and opportunism. Several aspects have contributed to the current

implementation of programmatic assessment. These include a clear view of the direction of changes to be made to convince the leadership and staff of the value of programmatic assessment; contributing to the programmatic assessment literature; distributing evidence from the literature amongst staff; participating in various decision-making bodies; and leading individual projects, such as the learning coach initiative. These contributions have made it possible to be agile in navigating a complex system where a linear approach to project planning proved ineffective.

Currently, we do not have systematic evaluation data on the effectiveness of the three elements of programmatic assessment (i.e., progress testing, the professional learning portfolio, and the learning coach initiative) in contributing to our students becoming better doctors. We do, however, have anecdotal evidence from students who self-report the development of new insights through continuous, deep reflection and analysis of their learning. These new insights are evident during learning coach meetings and through the professional learning portfolio, wherein there are noticeable shifts in the goal and motivation to study—from a focus on passing assessments toward learning to become a competent, safe doctor.

### 7. Lessons Learned

In reflecting on the implementation of PAL and the introduction of the learning coach initiative, we have had successes and identified opportunities that have required us to adapt our thinking. In this final section, we present key lessons we have learned in the process of implementing a fundamental educational change.

Given the complexities of embedding a programmatic approach within an existing university structure that emphasizes more modularized approaches to teaching and learning, compromises have had to be made in the process of implementing PAL. This has, in some instances, prompted the development of alternative and creative narratives in seeking to align with university regulations and policies. This has required a careful balance between strict adherence to, and working in the spirit of, policies.

Since the implementation of PAL in the MD program, the university context has changed. One recent major change has been the implementation of a new university assessment policy and procedures, which are in full alignment with assessment for learning principles and, to an extent, programmatic assessment. In the process of revising the assessment policy, the university has looked to the MD program for advice, and some of our approaches now serve as exemplars for the broader university community. In this case, we have learnt to recognize that *enacting course-level change can be a precursor to broader organizational change, and challenging existing, dominant structures can benefit the broader community.*

The initial approach of preparing an extremely detailed project plan, complete with timelines, milestones, and deliverables, was ineffective. In hindsight, this is logical. Education is complex and so are educational organizations. Linear plans, therefore, do not provide sufficient flexibility or agility to adapt in response to feedback. In the case of the Flinders University MD program, the change process became easier and was facilitated once the decision was made to first introduce the learning coach initiative followed by a gradual introduction to programmatic assessment. This has taught us to *be agile and flexible in developing plans for educational change.*

Medical educators typically assume many roles (e.g., teacher, assessor, curriculum developer, evaluator, educational leader, researcher, and scholar) [25]. In cases where there is an impetus for educational change, this can lead to medical educators becoming the problem owner, project manager, and project expert. In such situations, medical educators bear the responsibility for all aspects of the change, which can lead to disengagement by the broader university community. This can place medical educators in a precarious position. If they fail to enact the educational change, they risk losing credibility within the organization. Identifying champions for change across the organization, coupled with a supportive leadership team that promotes the delegation of change management responsibilities to a

range of stakeholders [26], can help to offset these risks and serves to build a community that embraces change. Hence, we have learnt about the value and importance of *building a community of change agents and champions for change*.

When the original implementation plan for PAL was developed at Flinders University, the Australian Medical Council had not yet made reflective practice explicit as a fundamental component of medical education. Additionally, they had not yet indicated that programmatic or longitudinal approaches to assessment would be the preferred approach. By the time the pilot portfolio and coaching program had concluded, these aspects had become more explicit and helped to provide further impetus for educational change.

More recently, the COVID-19 pandemic has served as an important catalyst for further enhancing the approach to programmatic assessment and assessment for learning in the Flinders University MD program. This required us to rapidly adapt our thinking and approach whilst recognizing the perspectives of various stakeholders. From a student support perspective, learning coaches were called upon to provide additional support to their students as they rapidly moved to online learning, disrupting their learning routines and access to social support. This pivot to fully online education challenged the ways in which assessments were delivered. The state of development of programmatic assessment and assessment for learning at the time was perfectly positioned to prompt a fundamental decision regarding control and agency. Considerations that needed to be made at the time related to examinations and whether to administer these through online proctoring (control), or adopt a position of student agency and fairness, the latter of which aligned well with an assessment for learning philosophy [14,27,28]. The Flinders University MD leadership team opted for this latter approach, which has contributed to significant improvements in student learning and satisfaction.

What these external developments have taught us is that in educational change, it is important to *be strategically opportunistic*. By this, we mean that change agents should be cognizant of both internal and external developments that help to build the case for change. Incorporating these developments into change narratives can aid in presenting a convincing argument for change to the various stakeholders.

## 8. Successes and Threats

Among the successes of implementing PAL into the Flinders University MD program, the learning coach initiative is now fully integrated across the MD program and has been recognized as instrumental in supporting students' development as reflective practitioners throughout their medical education. Individual student meetings coupled with the professional learning portfolio provide learning coaches with a wealth of meaningful information about the student journey through the MD program. This not only enables the coach to tailor their support for individual students but also provides information about the student experience that can be used to contribute to the ongoing improvement of all aspects of the MD program. Regular, ongoing student/learning coach interactions also mean that students who are experiencing difficulties that impact their progress and development can be identified—and issues addressed—in a timely manner. Indeed, the learning coach is often the student's first point of contact when they encounter difficulties.

From an MD course culture perspective, the learning coach initiative provides an important contact point for coaches. Many of our learning coaches are academic staff or clinical supervisors. By virtue of these roles, staff contributions to the MD course may be limited to one or two curriculum areas. As the learning coach role is longitudinal, spanning four years, staff are well-positioned to develop a deeper, more holistic understanding of the MD curriculum and assessment program. This not only supports individuals' understanding of the MD course and the rationales underpinning it but also builds collective understanding, contributing to the culture of the MD course community.

Our learning coaches value their role in meaningfully contributing to students' development and typically cite this as a key factor in their decision to become a coach. Learning coaches have commented that the initiative is a key 'point of difference' that sets Flinders

University apart from many other Australian medical schools. Indeed, the initiative is increasingly being cited as a reason that applicants choose Flinders University to undertake their medical degree. More broadly, some medical schools are increasingly recognizing the value of coaching in supporting student learning and development and are turning to Flinders University MD staff for support and advice.

Amidst these successes, however, are identified threats. The learning coach initiative requires engagement with, and long-term investment from, learning coaches. This proves challenging in the context of job uncertainty, short-term contracts, and resourcing—issues that exist in both academia and health care. This, coupled with a relatively large student body, requires an extensive investment of time for learning coach recruitment, training, and ongoing support. Despite these challenges, our MD program is currently supported by over 100 learning coaches.

Given the number of people involved in the learning coach initiative, ensuring consistency in coaching quality is challenging. This, coupled with different expectations amongst students and learning coaches, can impact the student experience and their engagement with reflective practice and coaching. Similarly, learning coaches' personal views on programmatic assessment and reflective practice, particularly if they are not aligned with those of the MD program, can impede students' engagement with these aspects of the course. Furthermore, our recent research and evaluation work has identified that learning coaches' level of understanding of self-regulated learning can shape their coaching practices. Where coaches have minimal understanding, this is likely to constrain the level of benefit students derive from coaching. We have gone some way to addressing these challenges by enhancing our initial training for new learning coaches, complemented by ongoing training and a coaching community of practice built upon case-based discussion. These opportunities do, however, require an investment of time, which, for many coaches, can be difficult. Additionally, where coaches adopt a 'fixed' mindset about their coaching approach, promoting engagement in ongoing learning opportunities can prove challenging.

## 9. Conclusions

In this paper, we have presented our perspectives on the process of implementing programmatic assessment and assessment for learning in the Flinders University MD program. We have reflected on both the process and the various factors that needed to be considered not only in implementing this new assessment program but also in its ongoing development and quality improvement. As we have outlined, fostering a culture of programmatic assessment that supports learners to thrive through coaching has required compromise and adaptability. The current culture of both the MD program and the university has us well-positioned to continue to seek opportunities to make improvements to ensure that our medical students are well prepared to enter the medical workforce.

**Author Contributions:** All authors (S.M.K., L.W.T.S. and J.H.J.) contributed to the development and conceptualization of this case study. All authors have read and agreed to the published version of the manuscript.

**Funding:** This research received no external funding.

**Institutional Review Board Statement:** Not applicable.

**Informed Consent Statement:** Not applicable.

**Data Availability Statement:** Not applicable.

**Acknowledgments:** The authors acknowledge the contributions of past and present Flinders University medical course staff for their work in facilitating the change process. Additionally, the authors acknowledge and recognize the ongoing contributions of the 103 current learning coaches in the Flinders University MD program.

**Conflicts of Interest:** The authors are all current learning coaches in the Flinders University MD program. The first author is the coordinator of the Flinders University MD learning coach program, and the last author was the previous Flinders University MD learning coach program coordinator. Both the first and second authors are currently employed by Flinders University. Because of these respective roles, the authors acknowledge that their roles within the organization have influenced their interpretation of the case.

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
