# Peer review of "Embedding a Coaching Culture into Programmatic Assessment"

_education, doi:10.3390/educsci12040273_

Round 1

Reviewer 1 Report

Thank you for inviting me to read this article, which I enjoyed very much. I found it well-written and it followed a coherent structure. I particularly liked the way in which you were open about the difficulties that you encountered and the compromises that you were forced to make. This helps to make your article very useful for readers.

While I agree with many of the assertions that you make in the introduction, it would be helpful to add some more supporting references if possible. It would be helpful to understand a little more about how the university context has changed. How much of this was because of your implementation of a form of programmatic assessment and how much was coincidental? It would be helpful for readers to understand more about how you dealt with resistance at a university level, as this appears a common reason for reluctance to implement programmatic assessment.

You have painted a very positive picture of the learning coach initiative. It would be helpful to hear more about difficulties encountered by coaches or students, expanding a little on your very brief reference. You mentioned the need for further refinement of some aspects of your programmatic approach to assessment. It would be helpful to add a little clarity with some specific examples.

Reviewer 2 Report

This study describes a case study on the complexity of a transition towards programmatic assessment. It provides an interesting insight in the approach of programmatic assessment, but mainly from a procedural and organisational perspective. I would welcome more theoretical underpinning of the choices made, and how the findings relate to other studies on this topic. 

Below I share my comments and questions, which my help to strengthen the manuscript:

- In the introduction, the authors indicate the factors that hamper structural educational changes in universities, such as long-held traditions, values, and beliefs. On p. 2, line 55/56 it is stated that three factors – external drivers, new insights, and changing societal needs – have all contributed to the implementation of programmatic assessment around the world. Can it be explained in more detail what is meant by these factors mean and how they relate to programmatic assessment?

- What is the definition of programmatic assessment used in this context? To my knowledge, programmatic assessment is based on a number of principles. On p. 4, lines 96-109, three elements of the assessment program (progress test, e-portfolio and learning coaches). How do these three relate to the theoretical principles of programmatic assessment?    

- On p. 4 it becomes clear why a programmatic assessment approach was desirable, I think this could be addressed earlier.

- In section 5 three elements of the assessment program are described. These were already introduced on p. 3. Maybe it is good to prevent this repetition, and to restructure some sections. For example, to first start why programmatic assessment is an interesting concept, why it could be an interesting answer to a number of deficits in current assessment programs, what do we know about the complexity of implementing this educational innovation and how this knowledge is translated to the programmatic assessment approach in this specific context (describing the three elements).

- The title of section 5 includes PAL, what does this abbreviation mean?

- P. 5 line 201: students are expected to analyse their results after each progress test. What evidence do the authors have that underlines that student are able to do this?

- A progress test is norm-referenced. How makes this individual analysis possible? And how does a progress test fit into the principles of programmatic assessment since students receive a grade?

- What is meant by: “The Progress Test represents ‘pixels’ that contribute to the whole image of the student’s development (p. 5, line 210)."

- P. 5, line 204: students of year 3 en 4 receive strategic study coaching when the result is doubtful of unsatisfactory. Why is coaching not provided to year 1 and 2 students?

- Section 5.2: how do students know which information in the portfolio will be used for assessment? And how are they supported in using the portfolio? It would be helpful to have some more specific insight in how students work on their portfolio.

- P. 6 line 258: “Coaching occurs at regular intervals (i.e., once per semester) throughout the four-year”. I wonder if this is enough and effective? What happens if students need much more coaching based on their learning process?

- How does the coaching support students in how information (pixels) will be used? Is it transparent for student which pixels will be used for summative assessment?

- In the evaluation on p.7 it is stated that still many assessments serve as summative. What does this say about the understanding of the concept programmatic assessment?

- The lessons learned are not surprising and in line with many other evaluations on programmatic assessment. Is it possible to give more insight in the effects of the interventions? Do students for example become better medical professionals?

In summary I think that this study provides a nice narrative of an university that is exploring the implementation of programmatic assessment, but that its contribution to the understanding of new knowledge on what programmatic assessment makes worthwhile the investment is quite small.
